# A Survey on RF and Microwave Doherty Power Amplifier for Mobile Handset Applications

**Maryam Sajedin** [1,2,*], **I.T.E. Elfergani** [1,*], **Jonathan Rodriguez** [1,2], **Raed Abd-Alhameed** [3] and **Monica Fernandez Barciela** [4]

1   Instituto de Telecomunicações, Campus Universitário de Santiago, 3810-193 Aveiro, Portugal; jonathan@av.it.pt
2   Departamento de Electrónica, Telecomunicações e Informática, Universidade de Aveiro, 3810-193 Aveiro, Portugal
3   Faculty of Engineering and Informatics, Bradford University, Bradford BD7 1DP, UK; r.a.a.abd@bradford.ac.uk
4   Department of Signal Theory and Communications, University of Vigo, 36310 Vigo, Spain; monica.barciela@uvigo.es
*   Correspondence: Maryam.sajedin@av.it.pt (M.S.); i.t.e.elfergani@av.it.pt (I.T.E.E.); Tel.: +351 218-418-454 (I.T.E.E.)

**Abstract:** This survey addresses the cutting-edge load modulation microwave and radio frequency power amplifiers for next-generation wireless communication standards. The basic operational principle of the Doherty amplifier and its defective behavior that has been originated by transistor characteristics will be presented. Moreover, advance design architectures for enhancing the Doherty power amplifier's performance in terms of higher efficiency and wider bandwidth characteristics, as well as the compact design techniques of Doherty amplifier that meets the requirements of legacy 5G handset applications, will be discussed.

**Keywords:** High power amplifiers; high efficiency; Doherty power amplifier; 4G; 5G; GaN-HEMT

## 1. Introduction

5G communications is an international initiative that aims to deliver next generation services that are power hungry and data intensive. To achieve the targeted 5G performance indicators will rely on phased-array MIMO (multiple-input and multiple output) antennas, new spectrum availability, and small cell technology, which in synergy aims to provide a communication platform to provide not only enhanced broadband connectivity, but to enable the Internet of things service coverage for smart manufacturing applications, and provide Ultra reliable and Low Latency services. This paradigm will put stringent design requirements on the system architecture in place, and beyond that on the RF front-end. A Radio Frequency Front-End (RFFE), as shown in Figure 1, as one of the key components of a mobile terminal, is powered by a low-voltage source, or even batteries, which has to cover a vast number of frequency bands in order to provide a high-level of integration [1]. In the design architecture of Figure 1, power amplifier modules combine multiplexers, filters, and RF switches blocks to provide highly integrated transmitter and receiver, which helps to reduce the manufacturer's time-to-launch. A key design requirement for power amplifiers (PAs) is energy efficiency at the required output power levels and the targeted operating frequency. This requirement is even more pronounced in 5G cellular networks to not only minimize operational expenditure, but also to reduce the carbon footprint that is associated with the PA lifecycle [2]. Moreover, in conventional RF front-end configuration, power amplifiers are optimized for a specific frequency band, which results in a narrow-band matching

scheme; therefore, the PA's operating range at higher frequencies is limited. On the other hand, spectrally efficient multi-carrier signal exhibits time-varying amplitude and phase characteristics due to wide and rapid variation of the instantaneous transmit power, resulting in a high peak-to-average ratio (PAPR) signal and wider occupied bandwidth [3]. The adoption of high PAPR modulated signals forces the power amplifier to operate at a large output back-off (OBO) to satisfy the stringent linearity requirements that are imposed by the wireless communication standards. This provides an amplifier device with 8 to 15 percent efficiency, and the implementation might be acceptable if the RF power requirements are very low [4]. As RF output power increases, power wastage can take significant cost, which translates into various forms, such as higher temperatures, more expensive heat transfer solutions and higher operating costs. Therefore, power amplifiers with higher back-off efficiency and linearity are required to enhance the overall transmitter performance.

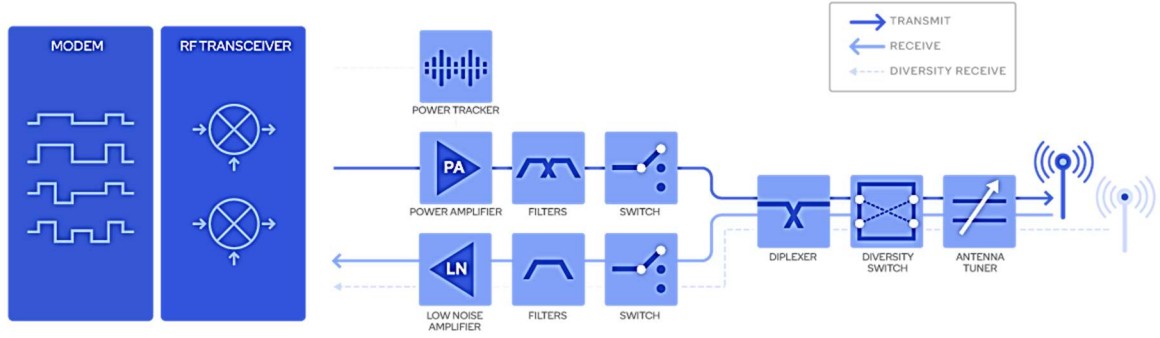

**Figure 1.** Qualcomm Radio Frequency Front-End (RFFE) solution [2].

Wide varieties of two-way power amplifier architectures have been introduced for efficiency enhancing without distorting linearity. In Envelope Tracking (ET) [5] and Envelope Elimination and Restoration (EER) techniques [6], based on the bias adaptation principle, the collector/drain supply of the RF power dynamically changes with the output envelope, thus the transistor operates with higher efficiency over a wide dynamic range of output power [7]. The ET PA can support modulation bandwidth of up to 80 MHz by utilizing digital predistortion [8]; however, the tracking bandwidth strongly relies on the supply modulator that needs the bandwidth enhancement for complex modulation signals. Several techniques have been applied to increase the bandwidth of modulator with penalty of complexity reflecting the additional circuit [9]. The EER technique generates constant envelope signal by changing the characteristics of high PAPR signal. One drawback of the EER/polar and ET amplifiers is the dependency of supply modulator performance on the amplifier efficiency, bandwidth, peak power, and dynamic range, which, in practice, severely restricts the instantaneous modulation bandwidth [10]. The operation of the Doherty Power Amplifier (DPA) that was originally proposed by W. H. Doherty in 1936 [11] and Outphasing by H. Chireix [12] are based on an active load modulation mechanism. The Outphasing architecture performs linear amplification by nonlinear components and it provides efficiency levels of 20–60%, and bandwidths of up to 40 MHz. In fact, the wideband Outphasing causes serious baseband overhead [13]. DPA operates in an optimal load impedance trajectory, which varies according to the amplitude of the input signal, which results in increasing the average efficiency of the Doherty PA without compromising its linearity. The DPA architecture provides an efficiency of 20–45% and bandwidths of up to 500 MHz [14].

The survey contribution is expressed in Section 2, we briefly address two main design challenges and potential strategies for 5G cm-wave/mm-wave DPA design, namely, the efficiency and bandwidth enhancement techniques. In this context, the most important characteristics of mobile handset power amplifiers, such as output power and power added efficiency, are targeted, which not only determine battery life, but also address the linearity/efficiency compromise in the handset amplifiers. An overview

of device technologies that are nowadays highly considered as promising technologies for the design of high efficiency, high power, and high linearity power amplifiers will be discussed in Section 3. Various design challenges of the RF and the microwave DPAs and effective solutions to overcome these issues have been introduced in Sections 4 and 5 respectively, along with the review on the recent research in the design and fabrication of the DPAs. Additionally, Section 6 discusses the bandwidth limiting factors and wideband design approaches of DPA. Next, the design methodologies for Multi-Band Doherty PAs will be introduced in Section 7. Then, the elaborated compact DPA circuit for handset applications will be discussed in Section 8. Finally, some conclusion will be given in Section 9.

## 2. The Survey Contribution

As a first contribution, this survey goes on to propose key developments of the Doherty topology to allow for exploiting complex modulation standards, when considering the available device technologies. Second, it suggests that practical solutions serve to overcome manufacturing limitations, which address the DPA-bandwidth degradation due to its contributing electrical components. A general significance of work comes up with looking to prior art for a deeper understanding of the design features challenges and applicable solutions for realizing energy-efficient, low cost, small size, and low complexity 5G mobile handset applications. These organization is important milestone towards building a superposition Doherty technique that can be deployed in supporting simultaneous transmission of multiple carriers that are formed by the carrier aggregation of non-contiguous spectrum.

## 3. The Choice of Device Technologies for RF-Front End Power Amplifiers

5G modern handset power amplifiers require lower output power than those that are currently used in 4G LTE due to utilizing higher Cm-Wave/Mm-Wave carrier frequencies and massive MIMO [15] technology. Moreover, mobile devices will need to support a wider set of RF bands, enable reliable connectivity, and require longer battery lifetime and the efficient use of electrical energy. The operation frequency band and the output power are the determining factors in choosing the semiconductor technology for power amplifiers design. GaN inherently shows high efficiencies, resulting in a reduction in system power consumption and presenting fewer thermal management challenges, which could ultimately lead to improved battery life [16]. Moreover, GaN devices can be downsized in fabrication, leading to much higher impedances that are more convenient for broadband matching. GaAs HBT is widely used in low power mobile devices, since it requires a single supply voltage, which is deemed to be a positive feature in any application, where a battery supplies the circuit. GaAs pHEMTs delivers excellent bandwidth, linearity, and efficiency, as shown in Table 1, for devices under one watt with low battery voltage, and thus serves as a strong candidate to develop millimeter wave PAs above 20 GHz [17]. SiGe RF PAs for handsets have become used in billions of RF FEMs for 4G handsets and WLAN products [18]. The LDMOS and GaN HEMT device technologies are widely used in base-stations due to their strong linearity attributes, besides being low-cost. DPAs based on CMOS technologies are also investigated due to their capability for co-integration and flexibility.

**Table 1.** Performance comparison of common semiconductors [15–18].

| Tech. | Frequency (GHz) | Power (W) | Gain (dB) | PAE% |
|---|---|---|---|---|
| LDMOS | <3 | 300 | <15 | Up to 70 |
| CMOS | 2.4 | 0.2 | 18 | 45 |
| GaAs MESFET | 12 | 0.08 | 5 | 65 |
| HBT | 2–8 | 2 | 9 | 20 |
| | 24–26 | 2.2 | 5 | 42 |
| | 3.5–10 | 1.3 | 8 | 25 |
| P-HEMT | 8–14 | 3.5 | 8.4–14 | 40 |
| | 28 | 1.6 | 16 | 35 |

## 4. Design Challenges of Doherty Power Amplifier

The two-way Doherty power amplifier implements by Carrier and Peaking active device stages. As depicted in Figure 2, it consists of a power splitter to properly divide the input signal to the device gates, and power combining network, including an impedance inverter to sum in phase the signals that arise from the two active devices; and, an impedance transformer that was connected to the output load. Finally, the phase variation that was introduced by the impedance inverter is compensated at the input of peaking amplifier.

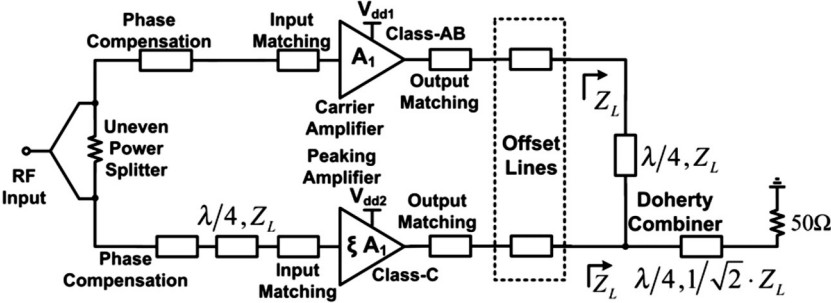

**Figure 2.** Two-way Doherty power amplifier scheme [19].

If the carrier and peaking transistors in the conventional architecture are represented by equivalent voltage controlled current sources, which are linearly proportional to the input signal voltage, the constant voltage at the carrier PA will be transformed into a constant current at the peaking PA, regardless of the load value. However, the practical challenges of a non-constant transconductance, non-ideal harmonics, knee-voltage, and effects of peaking amplifier's Class-C bias condition need to be addressed. However, the implementation of DPA presents other issues, such as nonlinearity at operating frequency, due to the device non-ideality. The Back-Off efficiency challenges of the Doherty PA are:

- Gain degradation: The peaking amplifier modulates the carrier one to deliver the maximum efficiency at the maximum power, this impedance variation and deviation from optimal load, results in a reduction of gain, Therefore, the individual transistors' gain response will be nonlinear.
- Phase distortion: the different conduction angles of the carrier and peaking devices result in non-similar output current profiles, which impose phase offset and gain imbalanced between two amplifiers stages [20]. Moreover, the parasites of real transistors will cause phase distortion and leads to a back-off efficiency drop and poor linearity.
- Poor inter-modulation distortion (IMD) performance: the peaking amplifier may cause a large distortion due to a low biasing condition (class C). One method of solving this issue is to deploy

the intermodulation products of carrier amplifier to add up with that of the peaking transistor at the load destructively to eliminate the IMD [21].

It should be noted that the instantaneous efficiency of power amplifier is a function of output power. In different classes of amplification, the instantaneous signal envelope can adapt the quiescent current. High quiescent current of Class A amplification causes the low IMD and low harmonic levels, which enable the amplifier to operate close to the maximum capability of transistor; however, the saturation voltage of the transistor deteriorates the efficiency [22]. Thus, Class A is typically used in applications with high gain and high linearity requirements. The quiescent current of Class B is fixed to minimize the crossover distortion at low output power, which enables linear and efficient amplification. In fact, the linear amplification refers to the short-circuited of all voltage harmonics of sinusoidal output signal. Increasing the load impedance, which provides larger voltage swing, can enhance the efficiency of this amplifier [23]. Class B is typically used in battery-operated, mobile radios, and base station amplifiers. By decreasing the conduction angle, the efficiency can be enhanced in the Class C mode. However, drive signal tends to increased, with the output power reduction, which results in low gain. Moreover, Class C mode is not often utilized in solid state amplification at high and microwave frequencies, because the reverse breakdown condition of transistor [24].

- Narrow bandwidth: The inherently narrow bandwidth behavior of DPA has originated from the quarter wave impedance inverter, which is usually applied for load modulation [25]. Moreover, the Doherty architecture integration into a single chip is a nightmare task due to the large size of quarter wavelength impedance network.
- Parallel parasitic losses: In the low power levels, the peaking device is in an open-circuit condition to avoid the current leakage to the carrier device. The traditionally adapted quarter-wavelength Impedance Inverter Network (IIN) can correctly perform load modulation only for real impedances in an ideal DPA [26]. However, the output parasitic reactances of real devices involve an imaginary part to the load, which must be eliminated. Furthermore, the output matching network can compensate one specific load impedance parasitic at saturation, which means that, for all other impedance introduced by load modulation, the reactive parasitics are not properly eliminated, and an unwanted phase rotation influences the load modulation [27], which results in lower back-off efficiency and DPA nonlinearity.

The common solution for restoring the optimum load-modulation is the insertion of two offset lines in the carrier and peaking output matching networks with characteristic impedance that is equal to the load impedance at saturation. An alternative method to compensate for parasitic effects is to integrate the device output matching networks and output combiner, which can reduce the device size [28]. Recently, the co-design method has been further improved while using the black-box technique at the output combiner, [29]. In this approach, the phase difference between two devices add more degree of freedom to achieve higher efficiency and extended the bandwidth.

## 5. Advanced Doherty Amplifier Architectures

The DPA high back-off efficiency can be achieved when considering some factors, including lower peak power level of carrier PA, than that of DPA, efficient operation of carrier PA at back-off, and small impedance matching loss. In this respect, the varieties of techniques have been proposed in order to restore the optimum active load modulation behavior. This section will present some of them.

### 5.1. Asymmetrical Doherty Power Amplifier (ADPA)

If the DPA implements by the same size of carrier and peaking amplifiers, the maximum output current of the peaking amplifier is smaller than that of the carrier amplifier due to the small conduction angle, which results in a reduction of the maximum output power. In fact, for an ideal behavior of the DPA, either the peaking device should be roughly double the size of the carrier device or the input

power splitting should be asymmetrical [30]. However, increasing the device size asymmetry causes severe gain degradation due to the stronger influence of the inherently lower gain Class-C peaking stage, which reduces the power that is delivered to the carrier stage and it affects the overall performance.

One of the most common techniques is using uneven power divider in favor of the peaking PA. Delivering more RF input power to the peaking PA, rather than to the carrier device, allows for the generation of sufficient current for the peaking PA to achieve a proper load modulation that increases the drain efficiency ranges from 10 to 13% [31]. Bias adapted DPA is depicted in Figure 3a, and it consists of two separate circuits, a fully matched Doherty amplifier, and an envelope shaped voltage generator for the carrier amplifier. Adaptive gate-bias control circuit continuously modifies the gate voltage of the carrier PA, which results in a reduction of the power consumption in the low power level when only the carrier device is operating. However, the main drawback of this solution is increased bias circuit complexity which requires additional cost to implement. Figure 3b shows a model of fabricated gate bias controlled DPA [32].

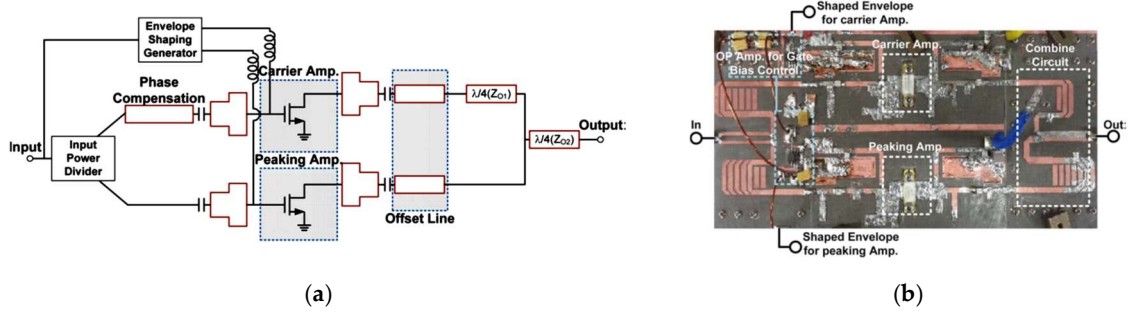

(**a**)　　　　　　　　　　　　　　　　　　　　　　　　　　(**b**)

**Figure 3.** (**a**) Schematic of the Doherty amplifier with gate-bias adaptation, (**b**) A model of fabricated Doherty Power Amplifier (DPA) and presented in [32].

Table 2 compares the performance of some ADPAs and Bias adapted DPAs in the literature. The results show that both the bias adapted-DPA and ADPA structures enhance the efficiency and output power characteristics with respect to the conventional DPA. The gain reduction of the ADPA in the high power levels results in a poor power-added efficiency. On the other hand, bias adapted-carrier PA ensures the full load modulation and the maximum output power.

**Table 2.** Measured performance summary of similar topology DPAs.

| Ref. | DPA Type | Frequency | Output Power | Efficiency (PAE) | Gain |
|------|----------|-----------|--------------|------------------|------|
| [30] | Gate bias Adaptation | 3.4–3.6 GHz | 49.3 dBm | 45% | 12.3 dB |
| [31] | Gate bias Adaptation | 1.94 GHz | 44.35 dBm | 60.5% | 12.75 dB |
| [32] | ADPA | 2.5 GHz | 47 dBm | 52% | 15 dB |
| [33] | ADPA | 2.14 GHz | 42 dBm | 48% | 15 dB |

### 5.2. Multiway and Multistage DPA

The conventional DPA is considered to be one of the promising approaches for improving efficiency over the 6 dB output power back-off. However, the power usage profile shows that, most of the time, the power amplifier operates at average transmitted output power in the range of 9–12 dB level below the maximum power; therefore, the 6 dB back-off efficiency improvement of the conventional DPA is insufficient and it results in the poor system efficiency [34]. It is possible to use more than two amplifiers, the so-called "Multistage DPA" configuration, to maintain the efficiency throughout the back-off region. The combination of the output power of several power amplifiers increases the

linearity (larger OBO). The Multistage DPA constitutes multiple Doherty PAs working in a parallel configuration, overcoming the issues that are related to the adoption of single very large device. In three-stage DPA that is shown in Figure 4a, the first peaking PA modulates the load of the carrier PA, and the second peaking PA modulates the load of the previous Doherty PA [35]. It can be observed that, in Figure 4b, the three-stage DPA provides additional peak in efficiency curve, which enables DPA for further enhancement on average efficiency.

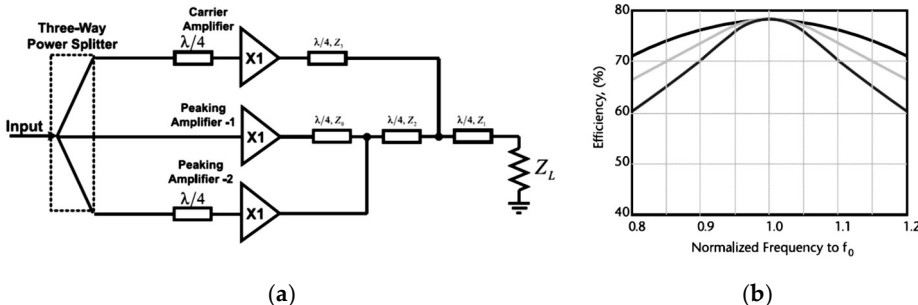

(a)                                    (b)

**Figure 4.** (**a**) Three-stage DPA architecture, (**b**) Efficiency versus frequency of improved three-stage DPA in [35].

The multi-stage DPAs can deliver highly efficient amplification of a modulated signal. However, these amplifiers are not very popular due to their complicated circuit structure, which poses problems in implementation. It is assumed that all of the carrier and peaking amplifiers reach their maximum current levels at the maximum output power, even though their biases are different. However, due to the lower supply voltage of peaking amplifiers, they cannot generate their respective full powers and the reduced output powers directly affect the efficiency, as well as linearity. Additionally, at the same current level, the smaller fundamental current levels of the peaking devices cause the modulated load impedances to become higher than those of the ideal operation, subsequently non-optimum load modulation. Table 3 compares the performance of some recent works on three-way and three-stage DPAs, which achieve reasonable compromise between the output power and efficiency without any further linearization method.

**Table 3.** Measured performance summary of similar topology DPAs.

| Ref. | Frequency | Output Power | Efficiency (PAE) | Gain |
|------|-----------|--------------|------------------|------|
| [34] | 2.14 GHz | 40 dBm | 35.2% | 9 dB |
| [35] | 3.5 GHz | 40 dBm | 37.3% | 11.1 dB |
| [36] | 2.5 GHz | 42 dBm | 30.85% | 23.84 dB |
| [37] | 2.14 GHz | 35 dBm | 39% | 10 dB |

*5.3. DPA Combined with the Envelope Tracking Technique*

Employing the input signal envelope tracking technique in the DPA is applicable for highly efficient handset PAs for operation in the low power region. In this amplifier, the gate bias of the peaking amplifier is controlled according to the magnitude of the envelope [38]. This technique can solve the common problem with the DPA, which is the fixed bias condition of the peaking PA. If both the peaking and carrier amplifiers adjust the gate bias, the overall system efficiency will be obtained by multiplying the efficiency of DPA with that of the supply modulator [39]. Envelope tracking can also be deployed on the carrier amplifier, leading to provide efficiency enhancement in low power level, then, while it goes to saturation in higher power and would bring about the modest efficiency. Figure 5a shows the block diagram of a DPA employing envelope tracking. In this architecture, the DC current for each PA decreases to a half, leading to larger impedance. First, the device is biased for operation in

low power levels, and then the bias is adjusted as the input power level increases making the peaking amplifier conduct more current to the load, while the carrier PA maintains the peak voltage. At the maximum power level, the peaking amplifier will have the same bias as the carrier amplifier, and its current contribution to the load will be equal to the carrier amplifier. Thus, the bias adaptation can help the peaking amplifier to compensate for the extra margin that it requires for synchronizing with the carrier amplifier. Figure 5b indicates The function of envelope shaping for the Doherty amplifier when the supply voltage is clipped to 3 V at back-off; thus, the extreme peak of envelope signal is reduced, and it results in an extension of dynamic range as well as high efficiency of the supply modulator [40].

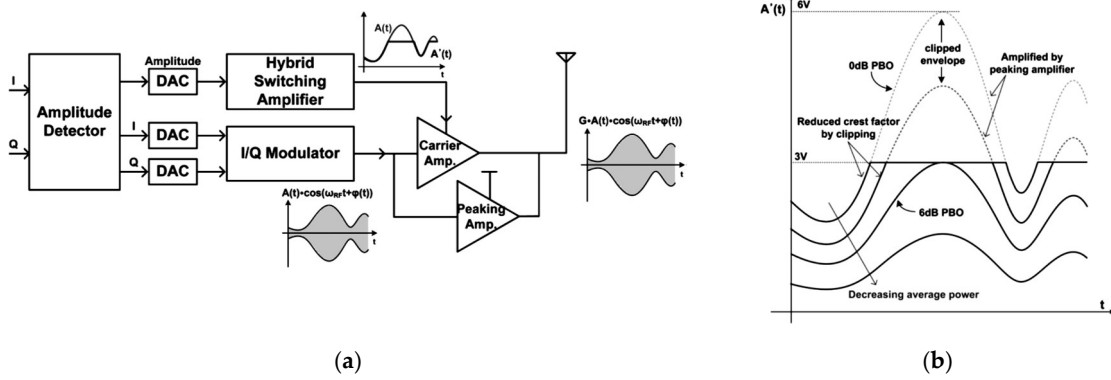

(**a**)　　　　　　　　　　　　　　　　　　　　　　　　　　　　(**b**)

**Figure 5.** (**a**) Block diagram of a Doherty PA with Envelope Tracking. (**b**) Carrier PA supply voltage at different output power levels [41].

This technique is attractive, because it can exactly synthesize the peaking amplifier current profile. However, providing tuning for ET-Doherty PA with three-port and additional circuit elements may increase the complexity and cost. Table 4 lists the published articles, which applied ET technology for the integration with DPAs for higher efficiency. The results show that using the supply modulation method reduces the gain reduction and provides a good PAE.

**Table 4.** Performance comparison of envelop tracking combined with Doherty amplifiers.

| Ref. | Signal | Frequency | Output Power | Efficiency (PAE) | ACLR (dBc) |
|------|--------|-----------|--------------|------------------|------------|
| [42] | WCDMA | 2.14 GHz | 38 dBm | 39.4% | −30 |
| [43] | WiMAX | 1 GHz | 37 dBm | 55.5% | −33.15 |
| [44] | WCDMA | 2.14 GHz | 42 dBm | 50.9% | −26.5 |
| [45] | LTE | 2.14 GHz | 38.4 dBm | 27.8% | −45 |

*5.4. Class-F Doherty PA*

Doherty can employ the switching mode for tuning two main harmonic components that enable amplifier to provide higher efficiency and power in order to restore the proper load modulation. Among the available techniques, the so-called saturated Class-F Doherty scheme has drawn the most attention because of its capability of outputting high power and providing high PAE performance [46]. The Class-F amplifier terminates the device output with the short-circuited of even harmonic frequencies of the fundamental component and open-circuited of odd harmonics to synthesize a perfect square waveform of drain voltage combination with half-sinusoid drain current [47]. Therefore, the overlap between the current and voltage waveform is near zero leading to reduce the dissipated power, the size, and weight of the power-amplifier. The second and third harmonic voltage components have the main contribution on Class-F design and higher order imply impractical suggest, because of circuit complexity. In fact, the harmonic generating mechanism of Class-F PA enables the Doherty PA to operate at quite high frequency (up to the X-band) for smart manufacturing applications [48]. A

number of practical aspects that are related to the finite number of harmonics, which can be efficiently controlled in actual devices, are addressed in [49]. In [50], the authors propose a new asymmetric Class-F$^{-1}$/F GaN Doherty while using Fourier transforms to compensate the low output current of peaking device, which mitigates the improper load modulation associated in conventional Doherty. The implemented device in this reference is depicted in Figure 6b. In other work [51], a blended Class-EF mode and load-pull technique for fundamental-frequency are proposed. The fabricated DPA delivers acceptable peak output power of 40.4 dBm at 84.4% drain efficiency. In addition, many linearization techniques have been adapted to variable envelope systems to overcome the nonlinearity issue associated with Class-F DPA, due to operating in saturation mode [52]. Figure 6a presents the schematic of Class-F DPA, where the wave-shaping network includes the harmonic control parts and the fundamental matching network.

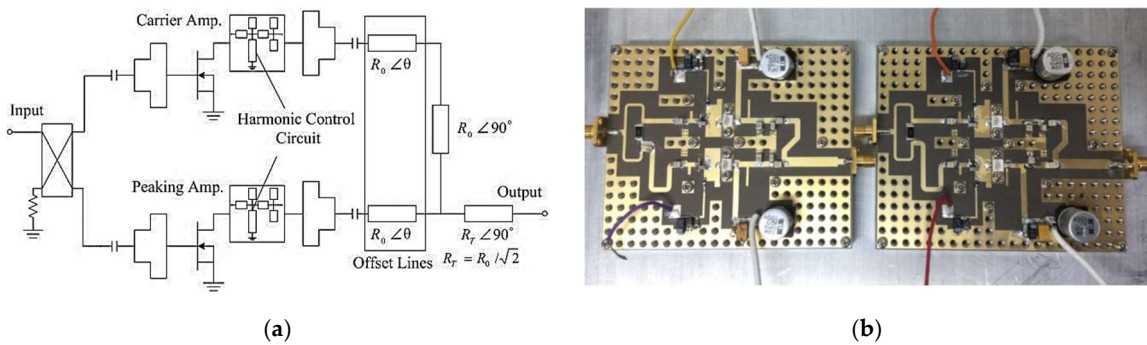

(**a**)　　　　　　　　　　　　　　　　　　　　　　(**b**)

**Figure 6.** (**a**) Schematic of Class-F based DPA [49], (**b**) Realized Class F$^{-1}$/F$^{-1}$ left, and Class F$^{-1}$/F right DPA in [50].

Table 5 indicates the employed technology, center frequency, output power, and efficiency of key works on DPAs employing the class F mode. It shows that the theoretical efficiency improvement can be achieved by providing the proper output harmonic loading conditions.

**Table 5.** Performance comparison of Class-F$^{-1}$/F Doherty Amplifiers.

| Ref. | Tech. | Frequency | Output Power | Efficiency (PAE) | Signal | Gain |
|---|---|---|---|---|---|---|
| [47] | GaN HEMT | 3.5 GHz | 40 dBm | 37.7% | WiMAX | 8.8 dB |
| [48] | GaN HEMT | 2.4 GHz | 44 dBm | 70% | LTE | 15 dB |
| [49] | GaN HEMT | 2.4 GHz | 37.3 dBm | 68% | LTE-Advanced | 15 dB |
| [51] | GaN HEMT | 2.14 GHz | 43 dBm | 49% | WCDMA | 6.7 dB |

*5.5. Class E Doherty PA*

Class E PA has similar non-linearity to the class C PA, but higher PAE performance. The transistor in Class-E mode operates as a switch to summing up the DC and RF currents and charge the drain capacitance. In the optimal condition, the drain voltage tends to become zero, when the transistor turns on, leading to the elimination of the losses for charging the capacitor. In fact, the shunt capacitor minimizes the overlap between the current and the voltage waveforms in the circuit of Class-E power amplifier. Figure 7a displays the schematic of a Digital Doherty PA while using the Class-E topology. Assuming that both on-resistance switch in the carrier and peaking path vary from 1000 to 0.1 [52]. At the beginning, the input power is low, and the on-resistance of the peaking amplifier switch is infinity. With the increase in input power, the on-resistance of the carrier amplifier switch ranges from 1000 to 0.1, yield an increase of output power and efficiency [53]. When the on-resistor of the carrier PA reaches its minimum value of 0.1, the obtained efficiency is 96.9% and is close to the ideal value of

100% [54]; then, the load of the peaking PA decreases from infinity to 0.1 with a further increase of the input current. Another efficiency peak is reached when the on-resistor of the peak amplifier is also 0.1 and it achieves 98% efficiency. Unlike the conventional DPA, it is obvious that the digitally controlled Class-E based DPA can potentially deliver high efficiency at the maximum and back-off power levels. Figure 7b shows the photograph model of a realized Class-E DPA.

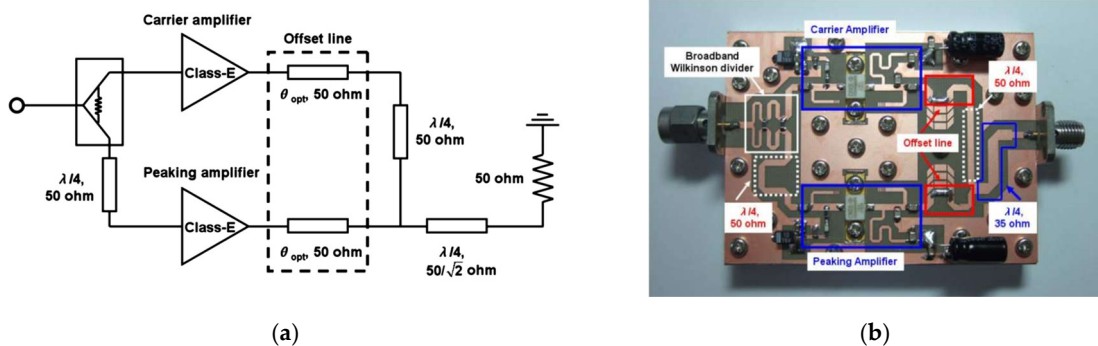

|       (a)       |       (b)       |

**Figure 7.** (**a**) Basic schematic of Class E based DPA. (**b**) Picture of Class-E DPA using GaN HEMT presented in [54].

Table 6 summarizes the performance of the Doherty power amplifier that is based on harmonic tuning; taking advantage of the soft-switching operation offered by Class E leads to high efficiency operation.

**Table 6.** Performance summary of Class-E Doherty Amplifiers operation

| Ref. | Tech. | Frequency | Output Power | Efficiency (PAE) | Signal | ACRL |
|------|-------|-----------|--------------|------------------|--------|------|
| [51] | GaN HEMT | 2.14 GHz | 43.1 dBm | 56.1% | WCDMA | −27.2 dBc |
| [52] | GaN HEMT | 2.85 GHz | 40 dBm | 42.9% | WCDMA | −26 dBc |
| [53] | GaN HEMT | 2.655 GHz | 42 dBm | 49.3% | WiMAX | −23.1 dB |
| [54] | GaN HEMT | 2.4 GHz | 45 dBm | 68% | CW | −26.6 dB |

The power matching circuits of amplifiers should be appropriately designed to cancel the IMD over power ranges across the wide bandwidth [55]. The theoretical load of 100 Ω (2*Ropt*) for the main amplifier is not actually the optimum load value, because of the knee effect of the transistor. The efficiency of traditional symmetrical DPAs will be enhanced by adopting an output impedance for the peaking device different from an open circuit and modifying the phase delay of the input matching network.

*5.6. Inverted Doherty PA*

In the inverted DPA structure, the inverter transmission line is connected to the drain of peaking PA instead of the carrier PA. A 25 ohm load is observed by the main PA to deliver maximum efficiency within the low power region by reversing the Doherty combining point. The impedance inversion in the conventional Doherty is accomplished with the 50 ohm λ/4 line, which is incorporated into the peaking output matching network, and the output is taken from the combining node. However, in inverted Doherty topology, the inversion λ/4 line is connected to the carrier amplifier and the offset line provides the off-state condition for the peaking PA, as depicted in Figure 8a, [56,57]. In fact, the measured output reflection coefficient that determines the lengths of the offset lines is rotated to a high impedance in the conventional DPA, while it is rotated to a low impedance for the inverted DPA. The inverted DPA provides the high efficiency at the low drive level; however, the key challenge of

the inverted DPA is designing the output matching networks of the carrier and peaking amplifiers. Figure 8b shows an example of fabricated 190W IDPA with the final stage in $14 \times 8$ cm$^2$ case [58].

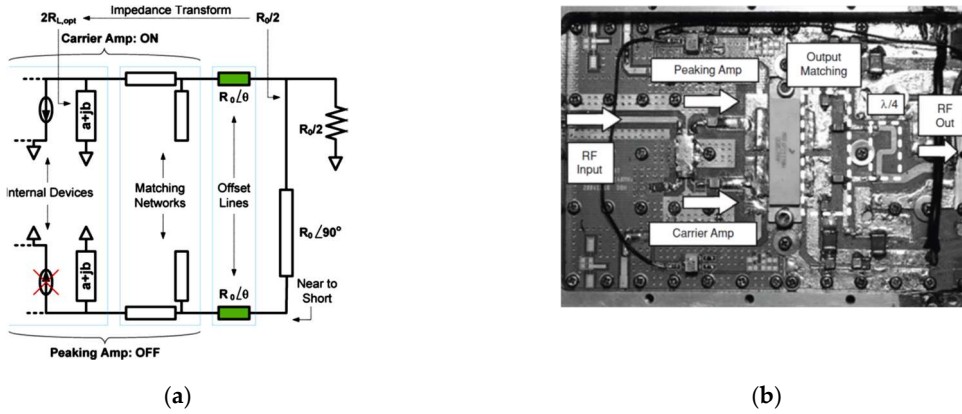

(**a**)                                               (**b**)

**Figure 8.** (**a**) Load network of inverted DPA operation [57], (**b**) Picture of the 190W IDPA described in [58].

### 5.7. Digital Doherty PA

An optimal load modulation behavior over a wide frequency band is achievable with good phase synchronization between the carrier and the peaking PAs. This can be obtained through reengineering the basic DPA structure, which leads to the so-called "digital DPA" [59,60]. This solution relies on a dual-input architecture for digitally assisted control over the input signals of amplifiers to mitigate the hardware impairments of the analogue circuit [61]. According to this strategy, the carrier and peaking PAs are individually driven from the baseband. Such a digital DPA eliminates the signal splitting device and minimizes the wasted energy of the driving power into the peaking path.

The Dual-Input DPA transmitter in Figure 9 consists of a DSP and a signal conversion block [62]. The DSP performs the modulation, interpolation, and digital pre-distortion (DPD) on an input signal. In this architecture, the DSP is applied to provide a main signal component along the carrier PA path, and a peaking signal component along the peaking PA path from a baseband input signal. The carrier and peaking PAs amplify the signals that are individually up-converted to an RF frequency. Precise alignment between the two paths at the baseband and isolation between the carrier and the peaking signals is crucial, which makes the implementation of the dual-input Doherty architecture challenging. Moreover, several limitations of digital DPA, like high oversampling, extensive pre-distortion requirements, and timing synchronization, restrict its application in multi-Gb $b/s$ modulations [63,64] and limit the modulation speed.

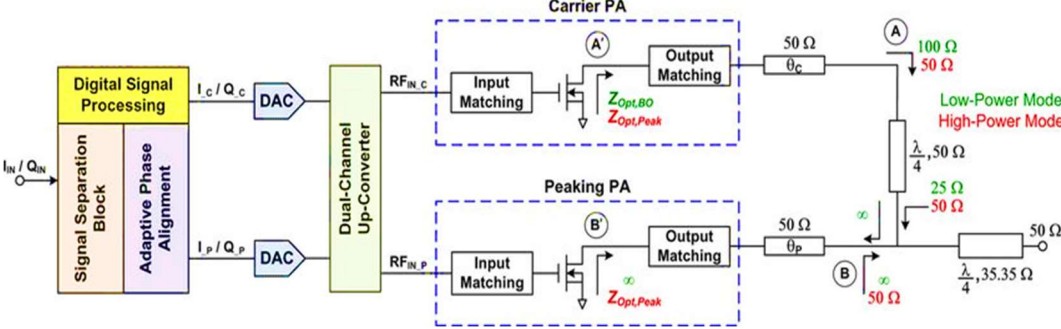

**Figure 9.** Block diagram of the dual-input digital Doherty PA architecture [62].

On the other hand, the analog PAs mostly show a significant distortion and gain compression. The paper work of [65] proposes an advanced hybrid use of mixed-signal Doherty-PA (MSDPA) in order

to overcome these limitations. In this work, the MSDPA combines analog carrier PA with a digital peaking one, which is driven by a complex modulated signal. Additionally, in this architecture, several peaking PAs are applied to extend the DPA compression point. The MSDPA extends the dynamic range by acting as an analog mm-wave PA in low power regime, in contrast to the digital DPAs with limited dynamic range due to the time-sampled interpolations of envelop signal. Besides, in high power levels, it works in mixed-mode to enhance the back-off efficiency by controlling the turning on instance of peaking PA. Finally, taking advantage of envelop-varying complex modulation at input, the MSDPA reduces the bandwidth expansion in mm-wave DPA.

Table 7 summarizes the recent research on RF and microwave DPAs. This table reports the employed technology, frequency target, output power, target OBO, and OBO efficiency. It is important to notice that all of the proposed techniques aim to enhance efficiency, gain, and linearity.

**Table 7.** Recent research publications for RF and microwave Doherty power amplifiers (PA).

| Ref. | Year | Technology | $f_o$ (GHz) | Power Gain | POUT (dBm) | PAE at 6 dB OBO | Signal |
|------|------|-----------|-------------|-----------|-----------|-----------------|--------|
| [66] | 2013 | SiGe HBT | 2.4 | 11 dB | 21 | 10% | LTE |
| [67] | 2017 | GaN HEMT | 2.14 | 10 dB | 35.5 | 39% | LTE |
| [68] | 2015 | GaN HEMT | 3.6 | 11 dB | 41 | 48% | LTE |
| [69] | 2017 | GaN HEMT | 1.8–2.2 | 12 dB | 50 | 50% | LTE |
| [70] | 2017 | GaN on SiC | 15 | 7 dB | 38 | 28% | WiMAX |
| [71] | 2014 | GaAs | 23–25 | 12.5 dB | 30.79 | 20% | WiMAX |
| [72] | 2015 | GaN HEMT | 10 | 9.2 dB | 36.02 | 47% | LTE |

*5.8. Linearization Techniques on Doherty PA*

An extensive research on the linearization techniques of Doherty PA has been performed in order to suppress its poor intermodulation distortion (IMD) performance. The most applied techniques include adaptive digital predistortion (DPD), post-distortion, feedback, and feedforward, as well as their combination. The regulatory agencies require the spectral emissions masked at close offsets (1.5 MHz) to be at least −45 dB, while, the nonlinear power amplifier exceeds this level by 15 to 20 dB, resulting in a significant distortion [73]. The linearizing techniques are focused on the harmonic cancellation of the power amplifier by optimizing the circuitry for modern applications. Various dedicated linearization techniques for DPA generate the second and forth order of the nonlinear signal at the output of peaking transistor through the band-pass filter, and feed into the carrier device with the fundamental signal, to suppress the third and fifth order intermodulation products [74]. An analog or digital splitter converts the input signal to dual-input independent branch signals in a typical predistortion approach, which can practically limit the efficiency and linearity performance due to a static function in Figure 10. Moreover, the conventional pre-distortion methods are restricted to low IMD cancellation, because the memory effect induces lower and upper spurious emissions [75].

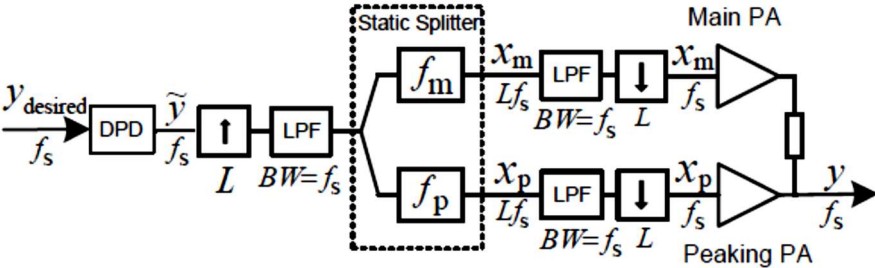

**Figure 10.** Linearization scheme in dual-input Doherty PA. $f_s$ is the initial sampling rate, L denotes the up sampling factor, $\chi_m$, $\chi_p$ are input signals of carrier and peaking PAs [74].

The dual-input Doherty PA, referred to as a digital DPA, generates two independent input RF signals to control the carrier and peaking PAs, leading to an extra degree of freedom that can be used to optimize its linearity and efficiency [76]. The varactor-based linearization architecture that was proposed in [77] adapted for dual-band DPA by utilizing additional input signal. In this method, if a proper envelope signal is injected, the transfer function of the DPA can be obtained by mapping function between the input and output signal and it can be inversed as a pre-distortion function. This scheme provides better performance in comparison with the single input linearization; however, it is not able to eliminate all of the intermodulation products of dual-input DPA. The linearized DPA commonly uses high sampling rates and fast digital to analog converters result in a spectral regrowth. In this respect, a digital predistortion method for a dual-input Doherty PA is introduced in [78], which performs the third-order Volterra-based function to drive the additional RF input signal to the peaking PA. Although, this dual-input DPD outperforms the single-input model in the reduction of the adjacent channel leakage power ratio, its outweigh complexity provides inadequate performance in high order nonlinearity. Figure 10 shows the block diagram of a conventional pre-distortion technique on Doherty PA in the digital domain.

The linearity of the high power Doherty amplifier can be improved by post-distortion-compensation, as proposed in [79], where the optimization is achieved by the cancellation of upper and lower symmetric sideband third order IMD products of devices by applying the two-tone input signal. In this work, a linearized transfer function of PA is modeled by the Taylor series function, when the gain expansion curve of the peaking device compensates the gain compression curve of the carrier PA, as can be seen in Figure 11. In another effort, a highly efficient feed-forward DPA is investigated [80] for base station application. This linear DPA optimizes the length of peaking compensation line and its gate bias by finding the output impedance of this device; however, it delivers a low PAE of 6–10%.

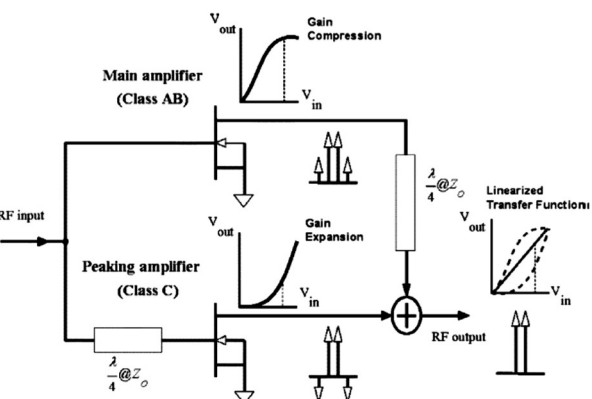

**Figure 11.** Linearization scheme of Doherty PA using post-distortion [79].

## 6. Doherty Bandwidth Extension

The efficiency enhancement of DPA was only maintained over narrow frequency bands and it offers a fractional bandwidth of smaller than 10% [81]. Although the conventional DPA is able to satisfy the modern handset requirements for efficiency, linearity, and output power, it is unable to meet the bandwidth requirement of the modern handset amplifier. Therefore, at present, the main challenge in DPA design is to extend its bandwidth. In conventional PAs, the output power or the gain define the bandwidth and the bandwidth range is achieved by the actual load power divided by the maximum power that could be delivered by the generator (available power). Beside, since DPAs are used to enhance the back-off efficiency, in this case a proper definition of bandwidth, is the frequencies range for which the back-off efficiency peak remains close to the maximum value achieved at the center frequency [82]. The DPA bandwidth increases with the increase in the input signal amplitude. While, the input power increases to provide the full voltage swing at the output, the bandwidth is not limited

at all and the DPA delivers 78.5% efficiency over the whole frequency range. Therefore, more attention has to be paid to the bandwidth behavior at the back-off power.

The degradation of the efficiency and output power of DPA from their maximum levels, over the entire range of frequencies can be observed in Figure 12a,b, respectively [83]. It can be noticed that the maximum output power is reduced by 25% with deviation from the center frequency. Therefore, the first efficiency peak does not actually represent an exact 6-dB back-off from the maximum output power, but actually higher than the 6-dB level. The main sources of frequency limitation that were typically observed in various implementations of the classical two-way DPA can be divided into two categories:

- Theoretical limitations: are directly related to the selected DPA topology and realize in the impedance inverter frequency dispersion, as well as the carrier and peaking transistors' current profiles.
- Practical limitations: attributed to the imperfections in the building blocks of the DPA (e.g., frequency dependence of the phase compensation network and offset lines, device non-idealities, and output/input matching networks).

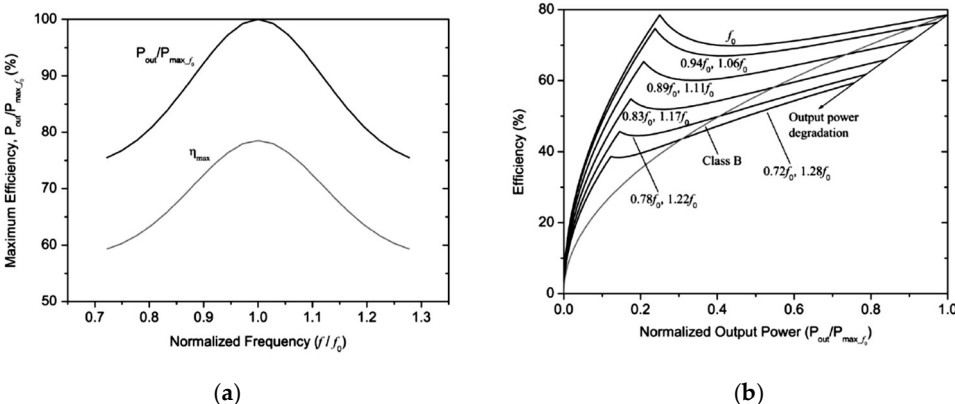

**(a)**            **(b)**

**Figure 12.** (**a**) Maximum efficiency and output power of DPA versus normalized frequency. (**b**) Efficiency of the DPA versus normalized output power at various frequency deviations from the center frequency [83].

The following sub-sections introduces the theoretical and practical bandwidth limiting components that contribute to constructing the DPA and present bandwidth extension techniques that have been proposed in recent research. Multiband capability in the conventional two-stage Doherty amplifier can be achieved when all of the Doherty constituent components are designed to deliver the corresponding performance over the required bandwidth of operation.

*6.1. Impedance Inverters*

When considering the carrier and peaking transistors as equivalent current sources in the two-way and three stages Doherty PAs in Figure 13a,b, respectively, the impedance that is seen by the carrier amplifier can be changed whilst the voltage swing across it has to remain constant, leading to maximized efficiency. Once the carrier amplifier saturates, the peaking amplifiers reduce the impedance that is seen at the output of the carrier PA by delivering current, which results in directing more current to the load, even when the carrier amplifier is saturated. Therefore, it is necessary to impose an impedance inverting network between the load and the carrier source, whereby the load impedance of each amplifier can be derived by the active load-pulling principle.

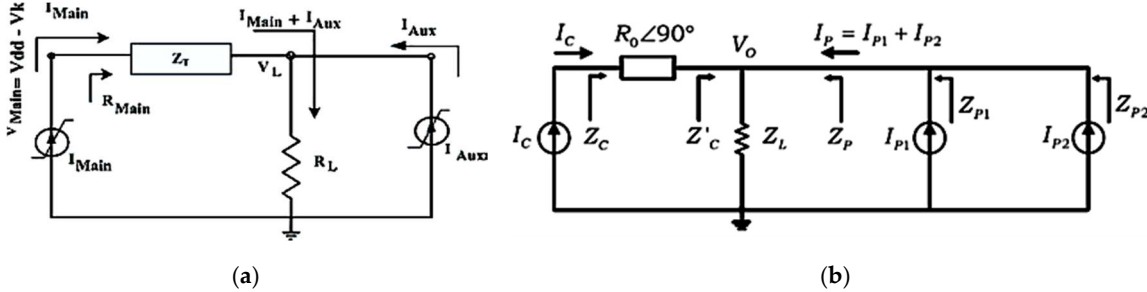

**Figure 13.** Structure of equivalent circuit of (**a**) Two-way, and (**b**) Three-stage DPA [84].

Impedance inverters, which are usually originated by means of a quarter-wave length transmission lines, provide perfect impedance inversion at the center frequency. By deviating from the center frequency, the length of impedance inverter has changed, and it has no longer $\lambda/4$ length, which shifts the purely resistive load that is seen by the carrier device (2*Ropt*) into a complex load disturbing the load modulation; that gets worse as the frequency deviation increases [85]. The frequency response analysis of the conventional DPA, in which the output combiner is realized by means of a $\lambda/4$ transformer, has been discussed in [86]. The optimum wideband operation of the DPA would require the transmission line (TL) to be a perfect impedance inverter over the whole desired frequency band, where not changing the physical properties of the TL is not possible; since the electrical length of the TL linearity increases with frequency. There have been various research works on the DPA to overcome the size constraint for handset applications, in which alternative impedance inverters, as surveyed in the following, replace the transmission line:

Three different impedance inverters, namely the coupled-line impedance inverter, short-circuit $\lambda/4$ TL compensated impedance inverter, and the open-circuit $\lambda/2$ TL compensated impedance inverter are proposed in [87], where the higher average efficiency could be obtained over a significant frequency range. The coupled-line impedance inverter consists of a coupled line section (CLS) and a single transmission line, where the electrical length of the coupled line and the feedback TL are both 90 degrees. This configuration provides the wideband impedance inversion properties that can be utilized to extend the bandwidth [88]. A short-circuit transmission line is the much simpler structure for a wideband impedance inverter by assuming that the coupling coefficient for the coupled lines is zero. However, the improvement is obtained by the expense of the bandwidth degradation at full power level for the coupled line impedance inverter and the open-circuit $\lambda/4$ TL compensated impedance inverter. Moreover, an additional $\lambda/2$ TL compensation line is connected to the quarter wave transmission line (QWTL) in the open-circuit $\lambda/2$ compensation transmission line, which can provide a wideband DPA, if the impedance of the compensation line is chosen to be equal to the optimum load impedance of the peaking device, and a proper input phase compensation network is applied.

Moreover, the effective load impedances of the Doherty amplifier can be expressed as a $[ABCD]$ parameters of the impedance transformer transistor, when the transistors are considered to be ideal current sources [89], as can be seen in Figure 14a. The $[ABCD]$ is the transmission matrix of the impedance transformer at a given frequency. In the low-power region, the effective resistance of the carrier PA has a maximum value of 2*Ropt* only at the center frequency, and it experiences a reduction as the frequency deviates. This leads to a back-off efficiency degradation as the frequency deviation increases, particularly when the DPA is operated over a wide bandwidth. The effective resistance of the carrier amplifier load impedance in the low-power region should be increased to extend the operation frequency band of the DPA. The resonance inductor-capacitor network (or *LC* tank) introduced in [90] is one of the possible solutions for reactance compensation. Additionally, a $\lambda 0/4$ short-circuited stub with a characteristic impedance of Z0 can extend the operation frequency, as illustrated in Figure 14b. However, the shunted *LC* tank or the short-circuited stub operations may affect the load impedances of both the carrier and peaking amplifiers at saturation, which can degrade the DPA performance.

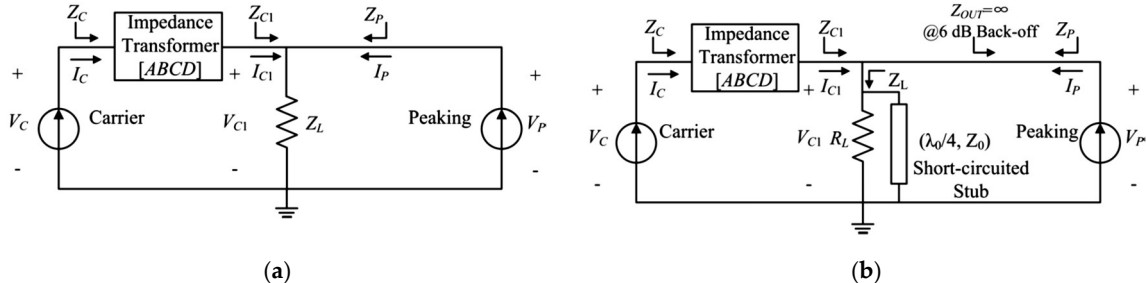

**Figure 14.** (**a**) Equivalent-circuit diagram of a DPA with the carrier and peaking amplifiers represented by current sources. (**b**) DPA using a shunted short-circuited stub [89].

*6.2. The Output Power Combiner Network*

The Doherty output combining circuit in Figure 15a consists of a quarter-wave transmission line with a characteristic impedance of 50 Ω to provide the required phase delay, and a quarter-wave transmission line of 35.35 Ω impedance that transforms the common load impedance to the final load impedance of 50 Ω. When the peaking amplifier is off, the output power combiner network acts as a 1:2 impedance transformer, which results in a loaded quality factor of 1.73 at 3 dB output-power that limits the broadband operation. In the high power levels, particularly at the maximum power point when the carrier and peaking amplifiers deliver equal amounts of output power, the combining circuit functions as a 1:1 combiner. With the maximum reflection coefficient $\Gamma_m$, the fractional bandwidth $\Delta f / f_0$ of the quarter-wave transmission line can be expressed by:

$$\Delta f / f_0 = 2 - 4/\pi cos^{-1}\left[\left(\frac{\Gamma_m}{\sqrt{1-\Gamma_m^2}}\right).\left(2\sqrt{Z_0 Z_L}/|Z_L - Z_0|\right)\right] \tag{1}$$

where $Z_L$ and $Z_0$ is the impedance pair for inversion [91]. Reducing the transformation ratio of the quartet-wave impedance inverter, making $Z_L$ and $Z_0$ closer together, can enhance the bandwidth of a quarter-wave TL. If the common load impedance is increased to a higher value, and then the transformation ratio of interconnection transmission line between peaking and carrier amplifiers will be reduced, which extends the bandwidth. Recent research proved that better bandwidth enlargement of the DPAs is obtained when the adopted output combiner circuit scheme is different from the conventional one. In [92], the authors proposed a DPA architecture for DPA center frequency, four sections of quarter-wave transmission lines are adopted instead of two, with different characteristic impedances, where two of the four characteristic impedances can be assumed as the free parameters; thus being usable to define the bandwidth of the DPA. This approach can optimize both the performance and frequency response of the DPA.

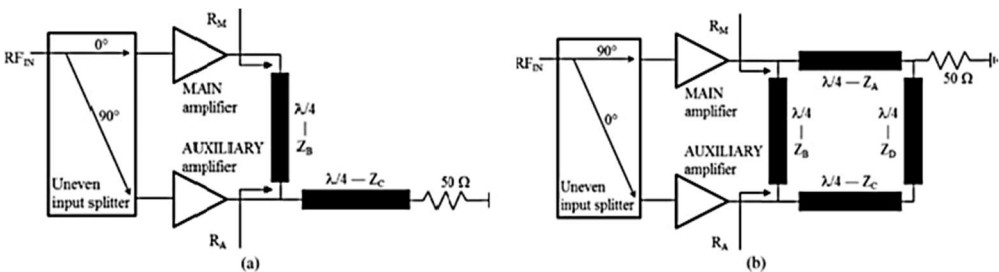

**Figure 15.** (**a**) Standard output power combiner and (**b**) four-section transmission line topology [92].

Quarter-wave TL stub can be replaced by an LC tank to reduce the size and broaden the bandwidth, as can be seen in Figure 16a. The resonator circuit provides open-circuit at the fundamental frequency and the short-circuit harmonics by large capacitor $(C_t)$, while a short-circuit quarter-wave stub behaves

differently at harmonics. It only provides short-circuit at even harmonics; therefore, the impedance of the carrier and peaking transistors are no longer the same at the maximum output power level [93]. In contrast, the Doherty combiner with a resonant tank holds the prime active load modulation of the carrier amplifier over the broadband. Figure 16b illustrates that the tank inductance $L_t$ acts as the biasing feed for both the carrier and peaking amplifiers to provide small baseband impedance at the drain.

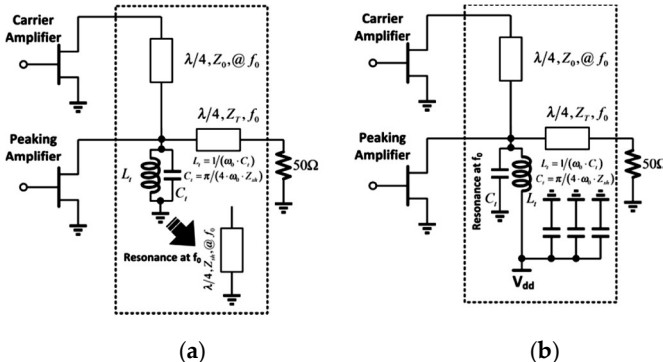

**(a)**                    **(b)**

**Figure 16.** The broadband output combiner employs a resonance *LC* tank at the output of the peaking PA (**a**) The short-circuited, quarter-wave stub as a resonator can be approximated by a parallel LC tank [93]. (**b**) The tank inductor Lt can act as the biasing feed for both the carrier and peaking amplifiers [94].

The digital assist Doherty mentioned in previous sections can be adopted, in which a pre-compensation mechanism acts on the input power distribution and the phase variation between the carrier and peaking amplifiers to compensate the frequency-selective behavior of the DPA to overcome the problem of bandwidth restrictions imposed by the quarter-wavelength transformers of the Doherty combining network [94]. The pre-compensation function will be dynamically adjusted based on the modification of current factors, which is used to model the variation of the output current of the carrier amplifier and that of the peaking device, respectively, in response to the changes in the injected input power at the center frequency.

*6.3. Offset Lines*

The offset lines are the essential components following the carrier and peaking amplifiers for optimum load modulation. The load impedance that was observed by the carrier PA at low power levels is twice $(2Z_0)$, and the output matching network is designed to match the optimum impedance determined in load pull and $2Z_0$, subsequently the efficiency and gain are maximized; whilst, at the output of the peaking amplifier, the offset line is adjusted to a high impedance. Thus, the offset line that was commonly used to compensate the output parasitic effects of the transistors and to modulate the output load impedances of both amplifiers to keep them close to the ideal values. Although this approach can be effective for single ended PAs, depending on the length of the offset line, as the operating frequency deviates from the center frequency, the output impedance of the peaking PA decreases. Thus, power leaks into this amplifier.

The offset lines and the impedance inverter in the DPA topology should be eliminated in order to enhance the bandwidth [95,96]. If the output capacitance of the PA device can be compensated in a more wide-band manner, the overall bandwidth of the DPA will be significantly improved. The linear output capacitance of a transistor can be absorbed by the quasi-lumped equivalent circuit of a quarter-wave line [97]. By doing so, the efficiency versus frequency improves, since, in this situation, the DPA is only limited by the bandwidth of the quasi-lumped transmission line. In [98], the authors eliminate the capacitive reactance by adding a parallel inductor at the device output capacitance, resulting in an *LC* resonant circuit that has a small bandwidth and it does not address the practical implementation of the DPA. Another attractive technique is to absorb the output capacitance along-with the connecting

bond-wires in the TL forming of the impedance inverter [99]. Figure 17a shows a wideband DPA with in-phase combination of both devices' output powers and Figure 17b shows the related equivalent schematic, which includes the output capacitance and connecting bond-wires at the active devices. Note that the transmission line along with bond wires and the output capacitance of the PA devices result in a quasi-lumped TL. Consequently, if the length and the characteristic impedance of this TL are properly adapted, it will act as an impedance inverter at the center frequency.

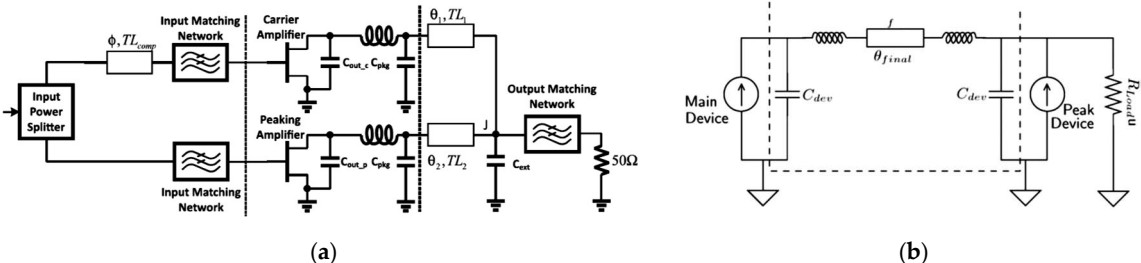

(**a**)    (**b**)

**Figure 17.** (**a**) Simplified scheme of broadband Doherty PA [95]. (**b**) Schematic of a DPA, which absorbs the device output capacitances and bond-wires into the quasi lumped transmission line impedance inverter [99].

## 6.4. Input Dividing Network

The DPA behavior, as a function of the input power, is usually divided in two regions: saturation and back-off. The Wilkinson power divider in Figure 18a is utilized in most of the Doherty designs to achieve power division and isolation. It consists of quarter-wavelength micro strip lines for output ports and resistors for isolation. Moreover, it is better than the direct power divider in terms of bandwidth, because the power division ratio is determined by the input reflection coefficient of each amplifier, resulting in higher efficiency. However, it is desirable to drive more power to the carrier at low power to prevent the peaking amplifier from turning on early, and it improves the gain and efficiency. Moreover, more power allocated to the peaking path at high power level leads to proper load modulation, desired power generation, and good linearity by IMDs cancellation.

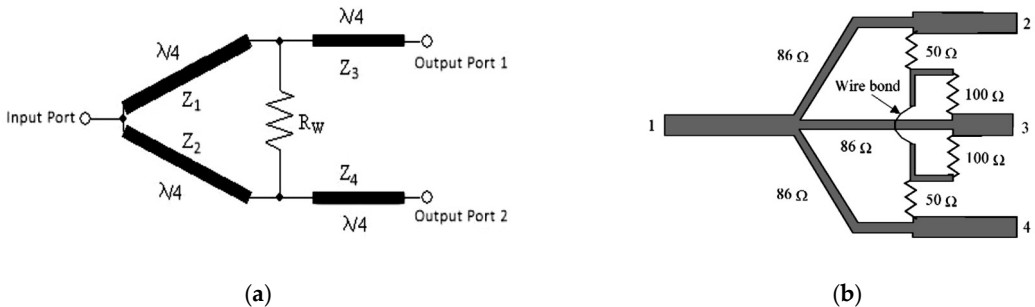

(**a**)    (**b**)

**Figure 18.** (**a**) Topology of unequal Wilkinson power splitter which uses a resistor that is normally open circuited (even mode) and does not generate loss (**b**) Compact micro-strip three-way Wilkinson power divider [100].

For a multiband operation with the center frequency ratio at each of the frequency bands of two or greater, the input divider can be configured as a multi-section Wilkinson power divider [100], which consists of stepped transmission-line sections with different characteristic impedances and electrical lengths. Figure 18b shows the compact micro-strip three-way Wilkinson power divider designed to operate over a frequency range of 1.7 to 2.1 GHz, with minimum combining efficiency of 93.8%, maximum amplitude imbalance of 0.35 dB, and isolation better than 15 dB in [101].

### 6.5. Phase Compensation Networks

The phase compensation networks are required to ensure the in-phase combination of the carrier and peaking amplifiers' output currents at saturation. The input and output matching networks may be not identical because of using different transistors based on bias condition and output power. Even with the same transistors, the matching networks may be different, since the two branches are working under different bias conditions, and they may have different input and optimum load impedances. Due to their different bias and matching conditions, the phase behaviors of the main and peaking amplifiers may not be identical over the bandwidth, consequently the phase compensation is difficult to achieve across the frequencies band while using a fixed-length offset line. A possible initiative to combat this challenge could be applying the high pass filter (HPF) with the same impedance transformation ratio in the input matching network of the amplifiers. This approach compensates the phase variation of the quarter-wavelength transformer and delay [102]. An alternative solution is to individually inject the input power to the amplifiers, so that the phase of the input signal can be adapted for each amplifier at each frequency [103]. However, this solution significantly increases the circuit complexity of the wideband DPA.

### 6.6. Input Matching Network

The appropriate impedance transformations at the transistor drain levels are realized while using impedance-matching networks. In the peaking PA, the input impedance increases as the input power increases leads to the gain expansion, whereas the input impedance is nearly constant and the gain compresses at high power for the carrier PA biased in class AB/B. Hence, the input powers delivered to the PAs should be properly adjusted to cancel gain expansion and compression. Proper input impedance matching is essential for highly efficient and linear operation since the input impedance of the peaking PA varies with the input power level using various matching circuits.

It is possible to use a multi-section matching transformer that consists of stepped transmission-line sections with different characteristic impedances and electrical lengths to provide an input broadband matching network [104]. Figure 19 shows the multi-section impedance transformer with N-1 sections matches in which the characteristic impedance of the transmission line are set to $Z_0$, to avoid impedance deviation in the matched bands. Such a structure is convenient in practical implementation, since there is no need to use any tuning capacitors.

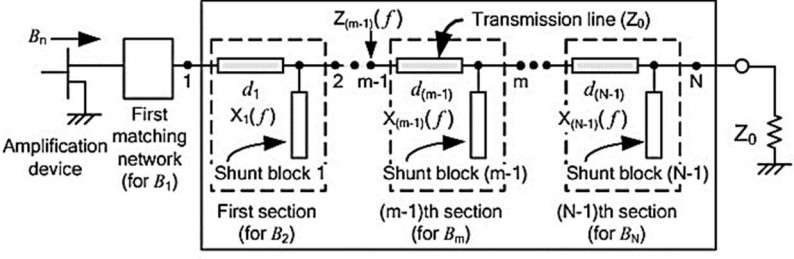

**Figure 19.** Basic topology of multi-section multi-band matching network [104].

### 6.7. Output Matching Networks

The bandwidth limitation arises when the load that is seen by the quarter wave transformer is unlike its characteristic impedance in the conventional DPA, where the load value matches at saturation, which is $R_{opt}$. This clearly means that in back-off, where the output load is $2R_{opt}$, impedance mismatch occurs. In a broadband PA design, the output matching network normally employs high-order topologies with filter structures to realize a certain impedance ratio in the operation band, which is necessary for high power amplifiers, because of the high impedance ratio [105]. Meanwhile, these high-order topologies provide an appropriate pass band with impedance ratio, and sufficient

out-of-band rejection to ensure the transmission of the power. Table 8 outlines the comparison of various reported wideband DPAs in the literature in terms of output power, fractional bandwidth, power added efficiency at maximum output power, and at 6 dB back-off output power in different frequency ranges.

**Table 8.** Performance summary of broadband Doherty Amplifiers.

| Ref. | Year | Technology | Frequency Range | Fractional Bandwidth | Output Power | Efficiency at Peak | Efficiency 6 dB OBO | Signal |
|------|------|-----------|-----------------|----------------------|--------------|--------------------|---------------------|--------|
| [106] | 2014 | GaN HEMT | 2 GHz | 36% | 37 dBm | 70% | 41% | WiMAX |
| [107] | 2019 | GaN HEMT | 480 MHz | 21% | 43 dBm | 63% | 50% | LTE |
| [108] | 2017 | InGaP-HBT | 800 MHz | 30% | 27.5 dBm | 45.3% | 41.2% | LTE |
| [109] | 2013 | GaN HEMT | 300 MHz | 14% | 42 dBm | 65% | 50% | WCDMA |
| [110] | 2016 | GaN HEMT | 1.1 GHz | 49% | 41 dBm | 57% | 50% | LTE |
| [111] | 2018 | GaN HEMT | 2.3 GHz | 87% | 43.01 dBm | 55% | 33% | WiMAX |
| [112] | 2018 | GaN HEMT | 1.1 GHz | 51% | 43.8–45.2 dBm | 56%–75.3% | 46.5%–63.5% | WCDMA |

## 7. Multiband Doherty Integration

The multi-band operation of a DPA is obtained through passive structures, such as impedance matching networks, phase compensation network, and impedance transformer network, due to their frequency dependency behavior. In fact, in the two close frequencies, these components introduce different phase relation between input and output signals [113]. In this respect, the multi-band Doherty constructing transmission lines are designed to work at the independent frequencies. The performance of dual-band is similar to single-band DPAs while taking advantage of choosing the operating bands, since the passive elements can be optimized for two operating bands at the same time. In [114], the authors propose a fully integrated 28/37/39 GHz multiband Doherty for massive MIMIO applications. The proposed power-award prototype is implemented in 0.13 µm SiGe BicMOS and it achieves high efficiency and extended carrier bandwidth. In this scheme, a compactness output network on-chip transformer to reduce the loaded quality factor and the impedance transmission ratio, which results in broadening the Doherty bandwidth, replaces the conventional output power combiner. Moreover, this adaptive feeding scheme dynamically modulates the peaking PA load impedance to increase its output current, leading to an enhancement of power gain. Some design approaches of dual-band passive components are discussed, as follows.

Dual-band impedance inverter functions at two uncorrelated frequency bands and it introduces phase shifting of $\mp 90°$ between its input and output ports. This condition is obtained when the real load resistance is transformed to the real resistance at the input port of impedance network. The equivalent impedance inverters at two arbitrary frequencies that are commonly realized by $T$ or $\pi$-network in Figure 20a, which are formed by two transmission lines with characteristic impedances of $Z_1$ and $Z_2$. The design equations for the T-network are given in [115]. In Figure 20a, the shunt stubs are 90° short-circuited and the line in the middle has electrical length of 180° at $f_0 = (f_1 + f_2)/2$. This inverter provides $\mp 90°$ phase shift at $f_1$ and $f_2$, where the output capacitance of the carrier and the peaking PAs can be integrated into the shunt stubs. Moreover, the shunt stubs improve the linearity of the circuit by serving as biasing feed. On the other hand, the quarter-wavelength line cannot provide multiband real to real impedance transformation operation due to its frequency dependency behavior. Therefore, cascaded transmission lines with different characteristic impedance that is introduced in [116], can provide a multipole response with different electrical lengths and different characteristic impedance ratio. The dual band two-section impedance transformation proposed in [117] is shown in Figure 20b.

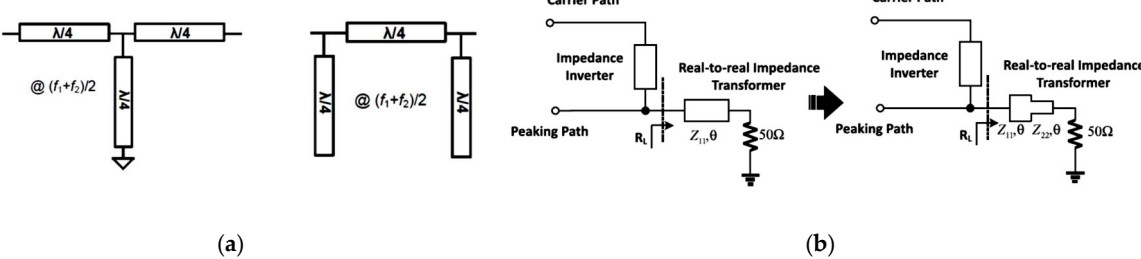

**Figure 20.** (**a**) Dual-band quarter-wave impedance inverter, T-network and $\pi$-network [115]. (**b**) A real-to-real impedance transformer is used to transform the output load to the required resistance at the Doherty common node: single-band and to dual-band [117].

The dual-band divider can provide equal power division for carrier and peaking PAs or send more power to the peaking PA and it can be configured as an in-phase multi-section Wilkinson divider or directional coupler. A suitable phase compensation network is required at the input to introduce the same phase shifting of the impedance inverter network. The branch-line coupler that is composed of four dual-band quarter wavelength transmission-line impedance-inverter can be based on $\pi$-section impedance transformers, or made up by four *T*-section quarter-wave transmission lines [118,119]. The phase compensation network can be directly absorbed into this component, providing appropriate output port connections.

The dual-band impedance matching network constructed at dual fundamental frequencies and second harmonics to ensure an appropriate gain and deliver maximum power. The dual-band phase offset lines are tuned for proper load modulation at the two frequencies with two arbitrary electric lengths, which leads to performance improvement. In Figure 21a, the Doherty load modulation circuit deploys the $\pi$-section network for dual-band implementations and Figure 21b shows the significant component on a sample of dual band DPA operating at 1.96 GHz and 3.5 GHz. The active load modulation varies with frequency due to different bias condition [120]. In fact, designing multi-band Doherty is critical, where all of the components require simultaneously addressing multiple band.

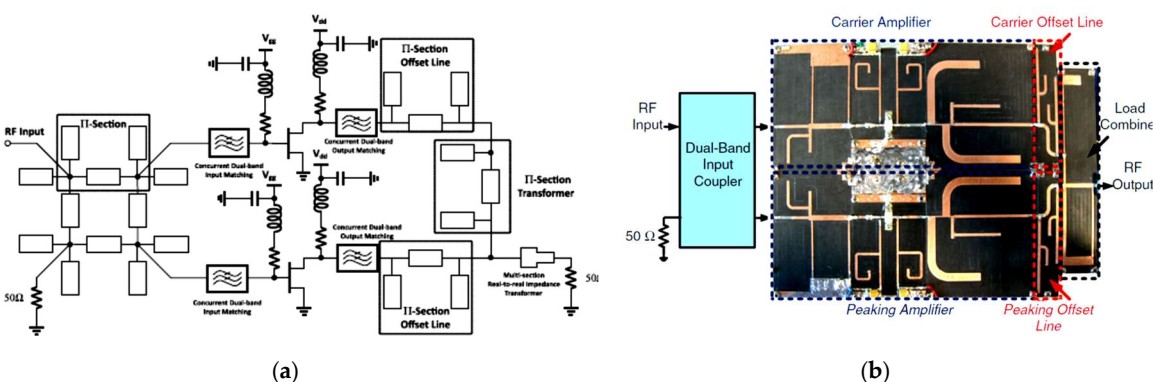

**Figure 21.** (**a**) Dual-band DPA architecture with $\Pi$-section structures [119]. (**b**) Dual-band DPA presented in [120].

The paper [121] presents the theoretical analysis and design technique of a load modulation balanced amplifier that is based on a specified output power and back-off level. In this architecture, the active matching network is performed by an adaptive output coupler, which controllers the reflection coefficient that is seen by amplifiers, Therefore, it does not need the bandwidth limited conventional output matching network. In this work, the design equations of output combiner are implemented based on an arbitrary black box network. The schematic of load modulation balanced amplifier is depicted in Figure 22, in which the Class-C control PA injects a signal to the output of the balanced amplifier to perform the load modulations, and the RF-input signal asymmetrically divided between

the balanced PA and controlled PA. Moreover, this paper compares the performance of single-frequency load modulation balanced amplifier with different load modulation architectures in terms of device periphery scaling between the carrier and control devices at the back-off power range for efficiency enhancement, and concluded that the conventional DPA and balanced PA show similar analysis of dynamic range, gain compression, and power division factor of splitter.

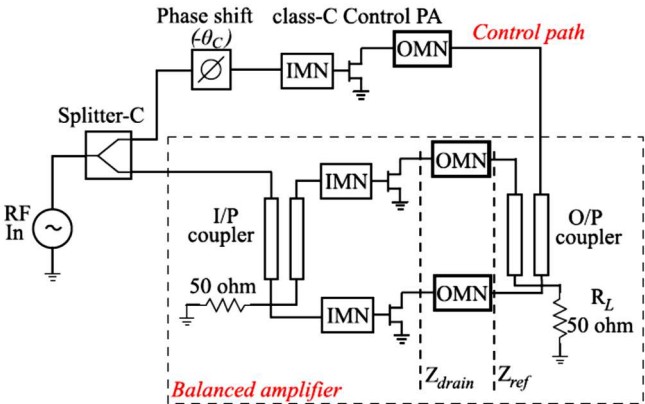

**Figure 22.** Simplified schematic of the RF-input load modulation balanced amplifier [121].

In [122], the authors develop the combination of Doherty and Chireix techniques in 2 GHz dual-input hybrid PA, in which the Outphasing angle is dynamically adapted with the incident power for proper load modulation. Recently, the Doherty-Chireix architecture has been investigated while using an output combiner with variable transmission lines in [123], and the realistic combiner when considering parasitics for wideband performance is discussed in [124]. The recent studies have shown that not only this continuum structure benefits from both Doherty and Chireix characteristics, but also it has the advantage of more design space to enhance the tradeoff between efficiency and linearity in comparison with single –input Chireix Outphasing PA mentioned in [125]. The Hybrid Chireix-Doherty (HCD) enhances the average efficiency of the modulated signal by reducing the efficiency drop between back-off and peak power observed in the conventional Doherty PAs. However, HCD complicated design increases the complexity and cost. In the mentioned work of [122], four fundamental current and voltage ratio factors of carrier and peaking devices generalize the HCD continuum theory, and the drain voltage of carrier device is set as equal to supply voltage at back-off and full power for higher efficiency. The implementation evaluation of HCD PA drain efficiency and back-off power range, proves that the HCD has a superiority performance in comparison with the Doherty PA for both the CW and modulated signals. Figure 23 shows the proposed HCD prototype in [122], where the carrier PA operates in Class-F mode of operation and peaking PA is biased in Class-C at the package reference plane. In this design, the two port output combiner is synthesized and implemented while using the Z-parameters, which is connected to the carrier and peaking PAs at two individually ports. Moreover, a stepped-impedance matching technique is applied for conjugate matching at fundamental frequency and the elimination of higher order harmonics. In the low power levels, the peaking PA is turned off in the Chireix-Doherty PA, similar to the Doherty PA.

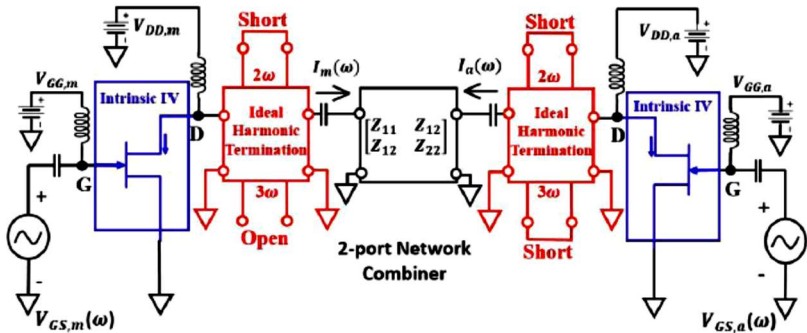

**Figure 23.** Hybrid Chireix-Doherty (HCD) PA prototype [122].

## 8. Compact Design of DPA for Handset Applications

The Doherty amplifier is less popular for handset application because of the size, bandwidth, and complex circuit topology. However, those problems can be solved while using compact design of the Doherty to integrate into a single chip [126]. By merging conventional components, such as power divider, the quartet-wavelength line for load modulation, and offset-lines for imaginary impedance modulation, the compact size and stable operation will be achieved. For mobile applications, the conventional bulky input power divider of Wilkinson splitter can be substituted by a direct power splitter to deliver more power to the gate of carrier amplifier in low power region due to its higher admittance, leading to compensate low gain, and peaking amplifier receives more power in high power region to increase the gain and output power [127]. The carrier amplifier could be matched to 50ohm for all power levels, because its impedance variation is small. Smaller size of the output circuit implementation can be achieved using inductance of the bias lines to resonate out the output capacitances of carrier and peaking amplifiers. Therefore, the output matching networks do not need to transfer the imaginary part of impedance. Moreover, the bulky quarter-wave inverter can be replaced by an equivalent lumped *LC* network for a Monolithic Microwave Integrated Circuit (MIMIC) handset application. Harmonic load condition and a compact design are the determining factors to choose appropriate lumped network type [128]. The lumped inverter of $\pi$-type shown in Figure 24a, reduces the size of the network since the two large capacitors provide short second harmonic load conditions at the output of carrier and peaking amplifiers; therefore, the second harmonic has a minor effect on linearity. Figure 24b illustrates the DPA implemented chip photograph with $1.1 \times 1.2$ mm$^2$ area size.

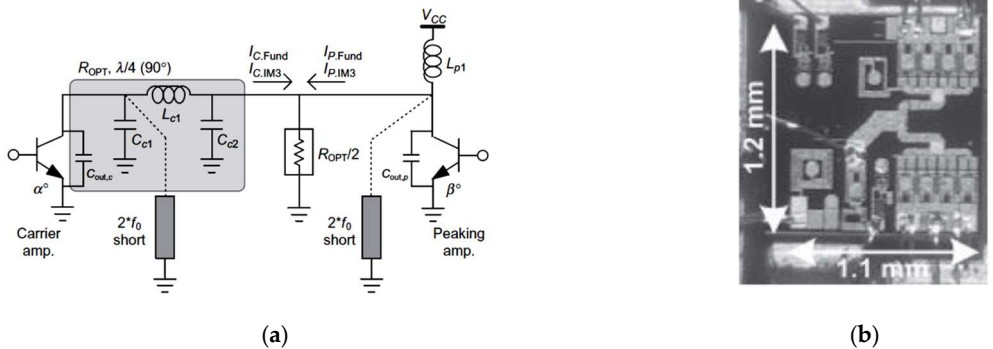

| (a) | (b) |

**Figure 24.** (**a**) Doherty amplifier using a lumped low-pass $\pi$-type quarter-wave inverter. (**b**) Picture of fabricated DPA using an InGaP/GaAs HBT described in [128].

The compact design of DPA employ lumped components with on-chip integration has been presented in literature [66,129] while applying different bias control techniques, including fixed, step, dynamic biasing, and logical. These techniques adjust the gate and drain biases as a function of the average output power to reduce the current and voltage at the low power level, resulting in a reduction

of power consumption. Figure 25a shows a block diagram of a MIMIC DPA while using an integrated bias adaptation circuit on chip with the same size of two-stages active devices and Figure 25b shows a sample of a module of printed circuit board as small as 1 mm × 1 mm [130]. At a low power level, the gate bias of carrier PA is fixed and tuning the drain bias according to the power level for high efficiency performs the load modulation. Once the same optimum impedance for both amplifiers are achieved at the higher power levels, the dynamic base bias control circuit increases the bias point of the peaking device to satisfy the linearity.

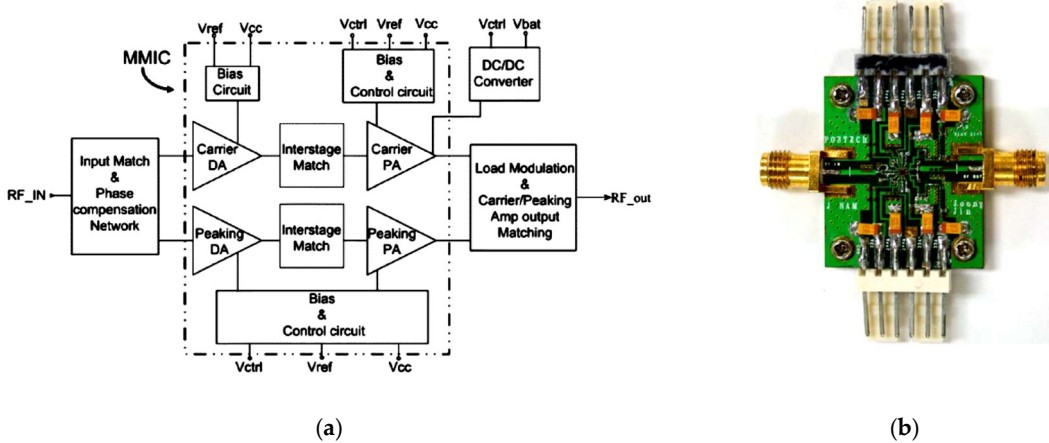

(**a**)                                                                 (**b**)

**Figure 25.** (**a**) MIMIC DPA using dynamic bias control. (**b**) A module of RF power test presented in [130].

A transformer-based DPA architecture is developed in standard 90 nm CMOS, adapting asymmetrical series combining transformer (SCT), allowing for chip implementation [131]. The SCT reduces the impedance matching loss in the silicon substrate by combination of multiple low voltage amplifiers. In this fully integrated un-even DPA architecture, by applying the proper number of Class-AB/C amplifiers according to the modulated envelope, a smooth transition for the whole power range and wide modulation bandwidth are achieved. It should be noted that the SCT is modeled using coupled lossy inductors, where the transformer ratio is the ratio of the carrier inductor to the total inductors. According to the implemented analysis of transformer based DPA, while using a very output large transformer ratio can improve the back-off efficiency of DPA, and it provides different size of carrier and peaking PAs to optimize the load modulation behavior. In fact, in a symmetrical transformer, the load seen by peaking PA is equal to the carrier PA at a high power level, while for asymmetrical transformer the load seen by peaking PA is smaller than that of carrier one, thus the peaking PA can deliver more power in comparison with carrier PA [132]. Therefore, in the un-even DPA, the peaking PA is optimized for the peak power operation. However, produced distortion by peaking amplifier with high transformer ratio can no longer be compensated, which degrades the linearity of DPA. Moreover, the fully integrated DPAs deliver low back-off efficiency, only because the carrier PA delivers power at back-off.

## 9. Conclusion

The Doherty PA is conceived as an efficiency enhancement technology for multimode multiband operation due to its low hardware complexity. Although it is ideally linear and highly efficient, the practical DPAs still suffer from nonlinear distortion and low average efficiency. Therefore, a variety of DPA circuit techniques have been adapted to improve the efficiency with extended back-off power, which includes the uneven power drive, gate bias adoption, saturated DPA, modified load modulation network, and digital DPA. Furthermore, in order to avoid the detrimental nonlinear effects of strong saturation, very accurate input splitting design, bias selection, and phase synchronization between two stages are required. Therefore, digital signal processing is often employed to adaptively adjust

these parameters. However, every approach that improves the PA efficiency usually has the inherent drawback of complexity. Among the cited advance architectures, DPA employing the envelope tracking technique not only minimizes the power gain reduction and total power dissipation, but also optimizes the efficient operation. Therefore, the ET-DPA is a promising candidate for high-efficiency amplification and an acceptable linearity; howbeit, the penalty of supply modulator complexity should be taken into the account.

On the other hand, DPAs are affected by several bandwidth limiting factors that pose broadband matching problem. This narrowband behavior of the DPA is mostly caused by traditional operating classes for active devices, conventional impedance inverters, offset-lines, and phase compensation network that makes it challenging to apply the DPA for multiband handset applications. A variety of referenced techniques have been carried out with the goal of improving the bandwidth of RF and microwave DPAs in both the analogue and digital domain. In fact, the digital DPAs provide better real-time configurability than the analog DPAs. However, the non-ideal behavior of electrical components and the limitations in fabricated techniques often narrow the number of practical solutions available. Embedding the offset-line within the output matching network, reducing the impedance transformation ratio, and eliminating the influence of output parasites are some introduced practical solutions, which serve to extend the bandwidth and to overcome the manufacturing limitations.

**Author Contributions:** Conceptualization, M.S. and M.F.B.; methodology, M.S.; formal analysis, M.S.; investigation, M.S.; resources, I.T.E.E. and J.R.; data curation, J.R.; writing—original draft preparation, M.S.; writing—review and editing, J.R. and I.T.E.E.; visualization, R.A.-A.; supervision, J.R.; project administration, J.R.

**Funding:** This project has received funding from the European Union's Horizon 2020 research and innovation program under grant agreement H2020-MSCA-ITN-2016 SECRET-722424. This work is also funded by the FCT/MEC through national funds and when applicable co-financed by the ERDF, under the PT2020 Partnership Agreement under the UID/EEA/50008/2019 project.

**Acknowledgments:** This work is supported by the European Union's Horizon 2020 Research and Innovation program under grant agreement H2020-MSCA-ITN-2016-SECRET-722424.

**Conflicts of Interest:** The authors declare no conflict of interest.

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
