# Peer review of "A Survey on RF and Microwave Doherty Power Amplifier for Mobile Handset Applications"

_electronics, doi:10.3390/electronics8060717_

Round 1

Reviewer 1 Report

- There are numerous typos and awkward English sentence, which makes many misleading statements.  A significant improvement on English writing is strongly recommended. 

- The manuscript is well organized. However, discussion on the important recent works (published within the last 3 years) are not sufficient. For example, the manuscript is missing discussion on multi-band Doherty PAs (Hu, ISSCC 2017), Doherty-like load-modulation PA (Pednekar, TMTT 2018), Doherty-Chireix continuum PAs (Liang, TMTT 2019), Doherty PAs for MIMO phased array (Noh, MOTL 2019), Mixed-signal Doherty PAs (Wang, ISSCC 2019), and so on. 

- Discussion on the usage of linearizers in Section 4.7 is very limited.  I would like to suggest the authors adding a brief summary of linearization techniques for Doherty PAs.

- Example performance numbers given in the Section 1 do not reflect the present state-of-the-art. For example, the authors mentioned that the envelope tracking PA bandwidth is up to 40 MHz.  This is not true anymore. Currently, the state-of-the-art envelope tracking PA bandwidths exceeds 80 MHz. (e.g., C.-Y. Ho et al., ISSCC 2018).

- Section 6 on compact design for handset applications is missing discussion on transformer based Doherty PAs, which is essential to the on-chip integration of Doherty PAs for mobile handsets.

- Most of the articles in the reference list are incorrectly formatted.  Many articles are missing publication title.

Author Response

Response to Reviewer #1:

 The manuscript is well organized. However, discussion on the important recent works (published within the last 3 years) are not sufficient. For example, the manuscript is missing discussion on multi-band Doherty PAs (Hu, ISSCC 2017), Doherty-like load-modulation PA (Pednekar, TMTT 2018), Doherty-Chireix continuum PAs (Liang, TMTT 2019), Doherty PAs for MIMO phased array (Noh, MOTL 2019), Mixed-signal Doherty PAs (Wang, ISSCC 2019), and so on. 

Response: The authors appreciate you for pointing out important modifications. We update the manuscript by adding “Multiband Doherty Integration” Section discussing very recent works on Doherty PA. This section includes the mentioned references in your comment: please refer to section 7, page 19-21.

 Discussion on the usage of linearizers in Section 4.7 is very limited.  I would like to suggest the authors adding a brief summary of linearization techniques for Doherty PAs.

Response: Thank you for your insightful comments. “Lineaarization techniques on Doherty PA” section has been added to manuscript discussing the most employed linearization techniques on DPA including Digital predistortion, Post-distortion and Feedforward techniques: please refer to subsection 5.9, parpagraph 1-3, page, 11-12.

Example performance numbers given in the Section 1 do not reflect the present state-of-the-art. For example, the authors mentioned that the envelope tracking PA bandwidth is up to 40 MHz.  This is not true anymore. Currently, the state-of-the-art envelope tracking PA bandwidths exceeds 80 MHz. (e.g., C.-Y. Ho et al., ISSCC 2018).

Response: This is a fair comment; thus, this statement is modified due to mentioned reference: please refer to the introduction section, paragraph 2. Page 2.

Section 6 on compact design for handset applications is missing discussion on transformer based Doherty PAs, which is essential to the on-chip integration of Doherty PAs for mobile handsets.

Response: Thank for this valid comment. We have extended this to be more consistent: please refer to section 8, paragraph 2, page 22-23

  Most of the articles in the reference list are incorrectly formatted.  Many articles are missing publication title.

There are numerous typos and awkward English sentence, which makes many misleading statements.  A significant improvement on English writing is strongly recommended. 

Response: We appreciate you for this important points. We have modified the ‘References’ according to the Journal format, also re-checked the manuscript for grammatical errors and improved it by English editing.

Reviewer 2 Report

The article provides a comprehensive survey on Doherty power amplifier. It will be better to include some pictures of actual circuits in terms of circuit size instead of all the schematics. Also, some discussion on the trade-off between different class PA would be useful.  

Author Response

The article provides a comprehensive survey on Doherty power amplifier. It will be better to include some pictures of actual circuits in terms of circuit size instead of all the schematics. Also, some discussion on the trade-off between different class PA would be useful.

Response: Thank you for your accurate comment. We have replaced some schematics by real devices, also the trade-off between efficiency and linearity in Class A, AB, B, C power amplifiers performance has been added to the revised manuscript: please refer to section 4, paragraph 3, page 4.

Round 2

Reviewer 1 Report

Thank you for the authors' efforts to revise the manuscript.  My previous comments were well addressed.  I have few minor comments.

1. This is a review paper, which surveys the Doherty power amplifiers but does not present any original ideas on a new power amplifier architecture.  Thus, to avoid confusion, I suggest the authors removing the phrase "As an original contribution to knowledge" in the beginning of the abstract.

2. Please do not use unnecessary capitalization. For instance, "Microwave" and "Radio Frequency" in the abstract should be corrected to "microwave" and "radio frequency".  Please revise the rest of the manuscript as well.

3. Some abbreviations were used without defining them (e.g., "DPA" in page 2).  The definitions should be spelled out for clarify (e.g., Doherty power amplifier (DPA)).  Please revise.

4. In Table 5 and 7, if GaN and GaN HEMT are meant for the same technology, please use one.

5. The left edge of the Figure 12 is clipped. Please correct.

6. Section 6.3 title has a spelling error.

Author Response

Response to Reviewer #1:

1.       This is a review paper, which surveys the Doherty power amplifiers but does not present any original ideas on a new power amplifier architecture.  Thus, to avoid confusion, I suggest the authors removing the phrase "As an original contribution to knowledge" in the beginning of the abstract.

We appreciate the reviewer for pointing out this important comment. To comply with this point, we have removed the confusing statement.

2.       Please do not use unnecessary capitalization. For instance, "Microwave" and "Radio Frequency" in the abstract should be corrected to "microwave" and "radio frequency".  Please revise the rest of the manuscript as well.

Thank you for your attention. We have modified the manuscript accordingly.

3.       Some abbreviations were used without defining them (e.g., "DPA" in page 2).  The definitions should be spelled out for clarify (e.g., Doherty power amplifier (DPA)).  Please revise.

We are appreciative that you have thoroughly read our manuscript. We have carefully revised and checked all abbreviations.

4.       In Table 5 and 7, if GaN and GaN HEMT are meant for the same technology, please use one.
We have reformed and rechecked the tables, according to your comment.

5.       The left edge of the Figure 12 is clipped. Please correct.
Thank you for your accurate comment. It has been changed for clarification.

6.        Section 6.3 title has a spelling error.

Thank you very much for your remark. The spelling error has been corrected in section 6.3.
